# Association of Vegetarian Diet with Chronic Kidney Disease

**DOI:** 10.3390/nu11020279

**Published:** 2019-01-27

**Authors:** Hao-Wen Liu, Wen-Hsin Tsai, Jia-Sin Liu, Ko-Lin Kuo

**Affiliations:** 1Department of Family Medicine, Taipei Tzu Chi Hospital, Buddhist Tzu-Chi Medical Foundation, New Taipei 231, Taiwan; kmichaelkkimo@gmail.com; 2Department of Pediatrics, Taipei Tzu Chi Hospital, Buddhist Tzu Chi Medical Foundation, New Taipei 231, Taiwan; tsaiwh.tw@gmail.com; 3School of Medicine, Tzu Chi University, Hualien 970, Taiwan; 4Division of Nephrology, Taipei Tzu Chi Hospital, Buddhist Tzu Chi Medical Foundation, New Taipei 231, Taiwan; sgazn.tw@gmail.com

**Keywords:** chronic kidney disease, dietary pattern, vegetarian diet

## Abstract

Chronic kidney disease (CKD) and its complications are major global public health issues. Vegetarian diets are associated with a more favorable profile of metabolic risk factors and lower blood pressure, but the protective effect in CKD is still unknown. We aim to assess the association between vegetarian diets and CKD. A cross-sectional study was based on subjects who received physical checkups at the Taipei Tzu Chi Hospital from 5 September 2005, to 31 December 2016. All subjects completed a questionnaire to assess their demographics, medical history, diet pattern, and lifestyles. The diet patterns were categorized into vegan, ovo-lacto vegetarian, or omnivore. CKD was defined as an estimated GFR <60 mL/min/1.73 m^2^ or the presence of proteinuria. We evaluated the association between vegetarian diets and CKD prevalence by using multivariate analysis. Our study recruited 55,113 subjects. CKD was significantly less common in the vegan group compared with the omnivore group (vegan 14.8%, ovo-lacto vegetarians 20%, and omnivores 16.2%, *P* < 0.001). The multivariable logistic regression analysis revealed that vegetarian diets including vegan and ovo-lacto vegetarian diets were possible protective factors [odds ratios = 0.87 (0.77–0.99), *P* = 0.041; 0.84 (0.78–0.90), *P* < 0.001]. Our study showed a strong negative association between vegetarian diets and prevalence of CKD. If such associations are causal, vegetarian diets could be helpful in reducing the occurrence of CKD.

## 1. Introduction

Chronic kidney disease (CKD) is global health problem with high risk of morbidity and mortality [1,2]. The prevalence of CKD is high and is gradually increasing [3,4]. Taiwan has a very high prevalence of CKD: 11.9% [5]. Risk factors for CKD include hypertension, diabetes, obesity, age, metabolic syndrome and taking nephrotoxins [6]. Based on the guidelines, management of CKD includes dietary and lifestyle modification [7]. A healthy diet may be associated with a lower incidence of CKD and a lower rate of CKD progression [8,9]. 

In recent years, several studies have shown that vegetarian diets are associated with lower prevalence of hypertension, diabetes, obesity, and metabolic syndrome [10,11,12,13,14]. These factors are the main causes of CKD. A study in Thailand revealed a significantly lower urinary protein level in vegetarians [15]. The literature has revealed that a plant-based diet also decreased the production of uremic toxins, inflammatory status, and oxidative stress [16,17,18]. These studies, which had small sample sizes, suggested that a plant-based diet can delay CKD progression, protect the endothelium, and decrease proteinuria [19,20]. According to these findings, several possible mechanisms could link the vegetarian diet to CKD. It seems reasonable to postulate that the renal consequences of hypertension, diabetes, and metabolic syndrome should be less prevalent in vegetarian populations. However, reports on the prevalence of CKD among vegetarians are currently still lacking.

To assess the evidence for the effects of vegetarian diets on CKD, we hypothesized that vegetarians, compared to omnivores, would have a lower prevalence of CKD. In this cross-sectional study, we aimed to investigate this hypothesis.

## 2. Materials and Methods 

### 2.1. Design and Study Participants

This study was a cross-sectional study to investigate the association between vegetarian diets and CKD. We analyzed individuals over 40-years old who received self-paid health exams at health checkup center of Taipei Tzu Chi Hospital (New Taipei City, Taiwan) between 5 September 2005 and 31 December 2016. We excluded individuals with incomplete information on demographic characteristics, alanine aminotransferase data, or serum creatinine data. The study was conducted in accordance with the Declaration of Helsinki and was approved by the institutional review board at the Taipei Tzu Chi Hospital (06-XD12-033). Informed written consent was waived because the study was a retrospective data analysis.

### 2.2. Clinical Assessment

The structured questionnaire, which included questions about sex, age, medical history, and health behavior, was conducted by a well-trained nurse. The dietary patterns were evaluated using a validated food questionnaire. We classified diets into several types (vegan, ovo-lacto vegetarian, or omnivore). An ovo-lacto vegetarian was defined as a person who consumes eggs or dairy products or both, but no other animal products; a vegan as one who only consumes plant-based foods; and an omnivore as one who consumes both plant- and animal-based foods. 

Height and weight were measured using an automatic electronic meter (SECA GM-1000, Seoul, Korea), and the body mass index (BMI, kg/m^2^) was calculated. Our well-trained nurse measured waist circumference (WC) at the mid-level between the lower edge of the rib cage and the iliac crest with the participants in a standing position. A WC of >90 cm in men and a WC of >80 cm in women was classified as abdominal obesity, defined by the HPA, MOHW, and the Taiwan definition [21]. An automatic blood pressure machine (Welch Allyn 53000, NJ, USA) was used to measure blood pressure (BP).

Venous blood was drawn after at least 8 hours of fasting. Measurements included serum total cholesterol (TCH), triglycerides (TG), and high-density lipoprotein cholesterol (HDL-C) (Dimension RXL Max integrated chemistry system, Siemens, Erlangen, Germany). Low HDL-C was defined as <40 mg/dL in men and <50 mg/dL in women. High TG was defined as ≥150 mg/dL [22]. Serum creatinine was measured using the alkaline picrate (Jaffe) method. The estimated glomerular filtration rate calculation was based on the Chronic Kidney Disease Epidemiology Collaboration (CKD-EPI) from serum creatinine [23]. 

Urine protein was determined by using single dipstick analysis with an automated urine analyzer (Arkray 4030, Tokyo, Japan). These results were reported as 6-grade scale: absent (less than 10 mg/dL), trace (+/−) (10 to 20 mg/dL), 1+ (30 mg/dL), 2+ (100 mg/dL), 3+ (300 mg/dL) or 4+ (1000 mg/dL). Patients with trace levels, 1+ level and above were defined as having proteinuria. The presence of CKD was defined as either presence of proteinuria or an estimated glomerular filtration rate (eGFR) ≤ 60 mL/min per 1.73 m^2^. 

### 2.3. Statistical Analysis

We compared three groups (vegan, ovo-lacto vegetarian, or omnivore) by chi-square and one-way ANOVA for the normal and continuous variables, respectively. When there were fewer than 5 observed values or the data did not conform to a categorical distribution, the Fisher’s exact test, and Kruskal-Wallis test, respectively, were used instead. For the incomplete cases in this study, we used the expectation-maximization (EM) algorithm to impute and to replace each missing value. The multiple imputation analysis for missing data was widely used for statistical analysis in many large cohort studies [24,25]. The multiply imputed data were analyzed using the multivariable logistic model to calculate the adjusted odds ratio (OR), and the approach to model selection used the stepwise backward and likelihood ratio test. Four separate logistic regression models were applied: an unadjusted model; a model adjusted for age and gender (Model 1); a model adjusted for age, gender, diabetes, and hypertension (Model 2); and a model adjusted for age, gender, diabetes, hypertension, abdominal obesity, systolic BP, low HDL, and high TG (Model 3). To confirm the results, we also subjected the dataset without the multiple imputation procedure to the same analysis and compared the results with multiply imputed dataset. The two-tailed test was used for statistical significance testing and a *p*-value of <0.05 was considered significant. All statistical analyses were carried out with SAS software version 9.4 (SAS Institute, Inc., Cary, NC, USA) and STATA 15.1 (StataCorp, College Station., TX, USA).

## 3. Results

### 3.1. Patient Characteristics

We initially enrolled 62,326 individuals, and total of 55,113 (45.8% men and 54.2% women) were included in final analysis after application of the exclusion criteria (Figure 1). The clinical characteristics of the individuals, divided according to the various diets, are shown in Table 1. Of these, 39,068 were omnivores, 4,236 were vegans, and 11,809 were ovo-lacto vegetarians. The mean age was higher (64.2 ± 9.9) for the vegan group than for the ovo-lacto vegetarian (60.4 ± 9.1) and omnivore groups (62.1 ± 10.1). 

The omnivores had a significantly higher prevalence of BMI, and higher percentage of abdominal obesity. Furthermore, the omnivores had significantly higher levels of total cholesterol and a higher percentage had high TGs, but this group was also significantly less likely to have low levels of HDL-C. (Table 1). 

### 3.2. Association of Vegetarian Diets with CKD

In our cohort, the mean eGFR was 84 mL/min per 1.73 m^2^ and the crude prevalence of CKD was 16.8%. A lower prevalence of proteinuria was seen in the vegan group. We found that the percentage of CKD was lower in the vegan subjects than in the other groups (vegan 14.8%, ovo-lacto vegetarians 20%, omnivores 16.2%, *P* < 0.001) (Table 1).

In logistic regression analysis using an unadjusted model, CKD was positively associated with age (OR 1.03, *P* < 0.001), male sex (OR 1.39, *P* < 0.001), current smoking (OR 1.86, *P* < 0.001), history of hypertension (OR 2.19, *P* < 0.001), diabetes (OR 2.53, *P* < 0.001), abdominal obesity (OR 1.37, *P* < 0.001), low HDL (OR 1.60, *P* < 0.001), high TG (OR 1.34, *P* < 0.001), and systolic BP (per 10 mmHg) (OR 1.15, *P* < 0.001) (Table 2). As shown in Table 2, vegan diet was negatively associated with CKD (OR, 0.90; 95% CI, 0.82 to 0.98; *P* = 0.017). However, the ovo-lacto vegetarian diet was positively associated with CKD (OR, 1.30; 95% CI, 1.23 to 1.37; *P* < 0.001). Four logistic regression models were also present in Table 2.

After adjusting age, gender, hypertension and diabetes (Model 2), subjects with vegan and ovo-lacto vegetarian diets were negatively associated with CKD. Further adjusting for sex, age, diabetes, hypertension, abdominal obesity, low HDL, high TG and systolic BP (per 10 mmHg) (Model 3), both vegan and ovo-lacto vegetarian diets were remained found to be significantly associated with CKD (vegan OR: 0.87, 95% CI, 0.75 to 0.97, *P* = 0.018; ovo-lacto vegetarian OR: 0.84, 95% CI 0.77–0.88, *P* < 0.001) (Table 2). 

## 4. Discussion

Previous studies have investigated the relationship between vegetarian diets and parameters associated with renal function [15,26,27,28]. Those findings suggested a significant effect on proteinuria but no significant effect on eGFR in vegetarians. However, the sample sizes in these studies were relatively small, and most of the subjects were Buddhist priests. The Buddhist priests lived an ascetic and communal life with little exercise. It was therefore difficult to extend the findings to the general population. Based on our best knowledge, this present study is the first large cross-sectional study to date to investigate the association between vegetarian diets and prevalence of CKD. 

Our study enrolled 55,113 subjects showed that only vegan diet was negatively associated with CKD compared to an omnivore diet in univariate analysis. After using the multivariate analysis, both two types of vegetarian diets (vegan and ovo-lacto vegetarian) were negatively associated with CKD compared to the omnivore diet. Diabetes and hypertension may contribute to the different ORs between the univariate and multivariate analysis. The comorbidity prevalence difference often existed in the diet patterns studies and sometimes confounder the relationship of the diet patterns and outcome diseases.

There are some special characters of our cohort. The participants in our cohort had a high mean eGFR (84 mL/min per 1.73 m^2^). This indicated that the CKD in our cohort was mainly early CKD with or without proteinuria. Moreover, the crude prevalence of CKD was accounted for 16.8%, which is slightly higher than the overall prevalence in Taiwan. In addition, the vegan group account for around 7.6% in all subjects, and the ovo-lacto vegetarian group account for 21.4%. The percentages of vegetarian diets were slightly lower while compared with Adventist Health Study-2 in the USA (vegan group: 8%, ovo-lacto vegetarian group: 30%) [29].This may be another special character of our cohort in Taiwan. 

Hypertension, diabetes, and metabolic syndrome are generally well-known as the main risk factors of CKD. Several studies have found that, after adjustments for age, sex, and body weight, BP is lower among vegetarians than omnivores [10,30,31]. A recent meta-analysis that included 39 studies with 21,915 subjects indicated that vegetarian diets were associated with lower mean systolic (−5.9 mmHg) and diastolic BP (−3.5 mmHg) [32]. Furthermore, vegetarian diets have a positive effect on body weight loss and are associated with a low rate of obesity [33,34]. The effect of the diet on BP cannot be explained by weight loss alone, and the dietary components of vegetarian diets include low salt intake, high potassium intake, rich fiber intake, protein intake from plant sources, each of which contributes to lowering BP [35,36,37,38]. 

Greater insulin sensitivity was found in vegetarians compared with omnivores [39]. Plant-based foods generally have low glycemic index values. The Adventists Health Study-2 showed that the possibility of developing diabetes was significantly lower in the vegetarian diet group compared with nonvegetarian group [40]. Unlike previous studies, subjects with vegetarian diets had higher prevalence of diabetes compared to the omnivores in our study. Many subjects in our cohort were self-disciplined Buddhist priests or volunteers, and they were prone to have more strict diet control with vegan when diagnosed with diabetes. This may explain why vegans had a higher proportion of diabetes in the vegan group. Due to the feature of cross-sectional study design, the cause-and-effect relationship between vegetarian diets and diabetes did not present in our cohort.

Many studies, including those from Western and Eastern populations, have demonstrated that a vegetarian diet is associated with a more favorable profile of metabolic risk factors and a decreased risk of metabolic syndrome [14,41]. Kurella’s study reported that metabolic syndrome is independently associated with an increased risk for the incidence of CKD among the nondiabetic adults [42]. In addition, the uremic toxin production rates of vegetarians are significantly lower compared with individuals on an omnivore diet [16]. These findings could reasonably explain why vegetarians have a lower prevalence of CKD compared to omnivores. 

Some limitations of this study must also be acknowledged. First, our subjects were selected from those undergoing health checkups in one medical center and may not be representative of the general population due to possible selection bias. However, the high percentage of vegetarians in this population provided us with an opportunity to evaluate the prevalence of CKD in subjects with different dietary patterns, especially vegetarian diet patterns. Second, there was no detailed information on portion sizes, the energy intake, or the nutrient composition of the food consumed by the participants. Thus, we could not analyze the role of the ingredients of the food. However, this was the largest study to explore the association of different vegetarian diets with CKD. Third, this study had a cross-sectional design, so a causal relationship could not be established between the vegetarian diet and CKD. Nevertheless, this is the first study to show an association between CKD prevalence and different vegetarian dietary patterns.

## 5. Conclusions

Our study demonstrated that a vegetarian diet was significantly associated with lower prevalence of CKD than in omnivores. If such associations are causal, vegetarian diets could be helpful in decreasing the occurrence of CKD and may be a potential recommendation for the prevention CKD. To confirm these data, more large-scale randomized controlled trials are needed.

## Figures and Tables

**Figure 1 nutrients-11-00279-f001:**
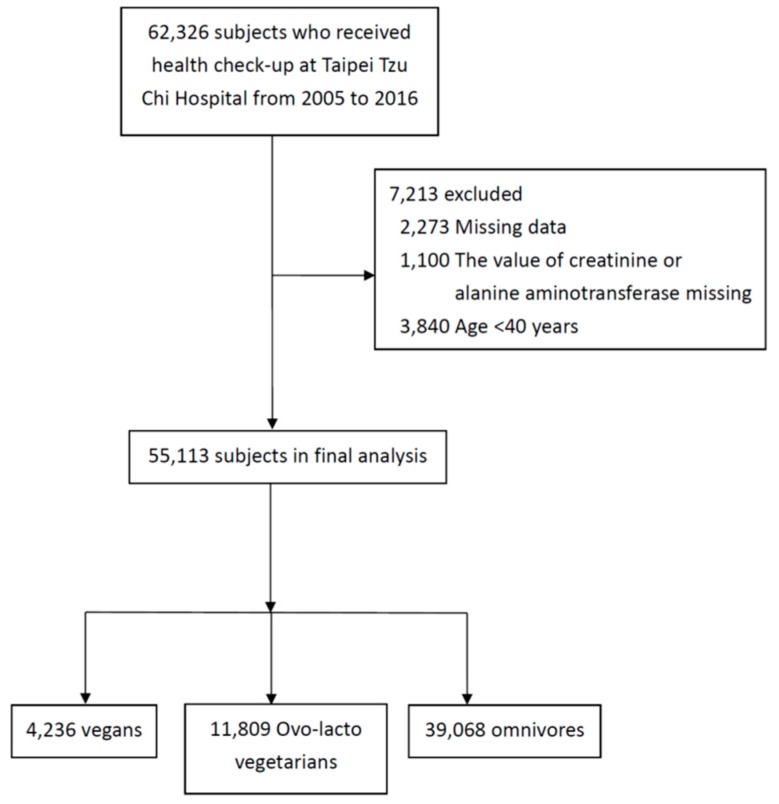
Subject selection.

**Table 1 nutrients-11-00279-t001:** Characteristics of individuals according to different diet patterns.

	Vegans	Ovo-Lacto Vegetarians	Omnivores	*p*-Value
*N*	4,236	11,809	39,068	
Age, years	64.2(9.9)	60.4(9.1)	62.1(10.1)	<0.001
Age group				<0.001
40–49, n (%)	240(5.7)	1098(9.3)	6,934(17.7)	
50–59, n (%)	963(22.7)	3305(28.0)	11,771(30.1)	
60–69, n (%)	1623(38.3)	4589(38.9)	12,542(32.1)	
≥70, n (%)	1410(33.3)	2817(23.9)	7821(20.0)	
Gender				<0.001
Male, n (%)	1412(33.3)	3966(33.6)	19,864(50.8)	
Female, n (%)	2824(66.7)	7843(66.4)	19,204(49.2)	
Comorbidity				
Diabetes, n (%)	269(9.4)	717(6.1)	2622(6.7)	0.042
Hypertension, n (%)	845(19.9)	2306(19.5)	7601(19.5)	0.074
Systolic BP (mmHg)	121(16)	120(15)	121(16)	<0.001
Diastolic BP(mmHg)	73(12)	72(12)	75(12)	<0.001
BMI (kg/m^2^)	23.1(3.3)	23.2(3.2)	24.1(17.5)	<0.001
Abdominal obesity, n (%)	56(1.3)	171(1.4)	1205(3.1)	<0.001
ALT (mg/dL)	25(18)	26(19)	32(27)	<0.001
Total cholesterol (mg/dL)	179(35)	183(36)	194(37)	<0.001
Low HDL, n (%)	1459(34.4)	4153(35.2)	9330(23.9)	<0.001
High TG, n (%)	957(22.6)	2345(19.9)	9840(25.2)	<0.001
Proteinuria, n (%)	114(2.7)	464(3.9)	1271(3.3)	<0.001
eGFR(CKD-EPI)	84(13)	84(13)	85(14)	<0.001
CKD	625(14.8)	2361(20.0)	6316(16.2)	<0.001

The data are shown as the number (%) or mean (SD). Abbreviations: BP, blood pressure; BMI, body mass index; Abdominal obesity was defined as a waist circumference of >90 cm in men; >80 cm in women; ALT, alanine aminotransferase; Low HDL was defined as HDL <40 mg/dL in men; <50 mg/dL in women; TG, triglycerides; High TG was defined as triglycerides ≥150 mg/dL; eGFR, estimated glomerular filtration rate; CKD, chronic kidney disease; Chronic kidney disease was defined as eGFR ≤60 mL/min per 1.73 m^2^ or proteinuria.

**Table 2 nutrients-11-00279-t002:** Logistic regression analysis of the presence of chronic kidney disease.

Variable	Unadjusted Model	Model 1 ^a^	Model 2 ^b^	Model 3 ^c^
	OR (95% CI)	OR (95% CI)	OR (95% CI)	OR (95% CI)
Age (years)	1.03 (1.03–1.03)	1.03 (1.03–1.03)	1.01 (1.011–1.02)	1.01 (1.01–1.02)
Gender (Male)	1.39 (1.33–1.45)	1.43 (1.36–1.49)	1.27 (1.19–1.36)	1.27 (1.19–1.35)
Diabetes	2.53 (2.35–2.73)		1.59 (1.42–1.78)	1.52 (1.36–1.69)
Hypertension	2.19 (2.09–2.31)		1.48 (1.37–1.60)	1.40 (1.30–1.52)
Abdominal obesity	1.37 (1.21–1.56)			1.01 (0.83–1.21)
Systolic BP (per 10 mmHg)	1.15 (1.13–1.16)			1.03 (1.01–1.05)
Low HDL	1.60 (1.53–1.68)			1.20 (1.12–1.28)
High TG	1.34 (1.28–1.41)			1.12 (1.04–1.21)
Omnivores	1.0 (reference)	1.0 (reference)	1.0 (reference)	1.0 (reference)
Vegan	0.90 (0.82–0.98)	0.83 (0.76–0.91)	0.85 (0.74–0.97)	0.86 (0.75–0.97)
Ovo-lacto vegetarian	1.30 (1.23–1.37)	1.31 (1.24–1.38)	0.83 (0.77–0.89)	0.82 (0.77–0.88)

Abbreviations: BP, blood pressure; TG, triglyceride. Abdominal obesity was defined as a waist circumference of >90 cm in men; >80 cm in women. Chronic kidney disease was defined as an eGFR ≤60 mL/min per 1.73 m^2^ or proteinuria. High TG was defined as triglycerides ≥150 mg/dL. Low HDL was defined HDL <40 mg/dL in men; <50 mg/dL in women. a: adjusted for age and gender; b: adjusted for age, gender, diabetes, and hypertension; c: age, gender, diabetes, hypertension, abdominal obesity, systolic BP, low HDL, and high TG.

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
