# Peer review of "Association of Vegetarian Diet with Chronic Kidney Disease"

_nutrients, 2019, doi:10.3390/nu11020279_

Round 1
Reviewer 1 Report
This retrospective study deals with the potential impact of dietary patterns on the occurrence of chronic kidney disease in humans.
The hypothesis and aim of the study are interesting, however, in my opinion, the results should be presented more carefully and differentiated.
Depending on the statistical test that was used, varying results were detected. In Table 1, it is demonstrated that the percentage of CKD was the lowest in vegans, but the highest in lacto-ovo-vegetarians. The authors should discuss this finding and differentiate that the occurrence of CKD was only low in the lacto-ovo-vegetarian group after adjustment of the data (lines 142). At the moment, the authors only state (in the Abstract, Discussion and Conclusions) that vegetarian diets were negatively associated with CKD, however, this is not completely true.
Moreover, the number of patients was markedly higher in the omnivorous group (n= 39,068) when compared to the ovo-lacto-vegetarian group (n=11,809) and the vegan group (n=4,236). Please comment on this, does it implicate any limitation for the statistical data analysis?
Minor comments:
Line 93: „three groups“ – please define
Line 102: please replace „were“ by „was“
Lines 111/112: please add the age for the omnivorous group
Line 156: „most of the subjects were Buddhists“ – please explain, why this is a limitation
Line 166: „causes“ – or do the authors mean „risk factors“?
Line 182-183: „In addition, a plant-based diet that focuses on fruits, vegetables and fiber is strongly believed to protect against the development of metabolic syndrome.“ – this aspect is already mentioned in the sentence in Lines 180-182 („…and a decreased risk of metabolic syndrome“) and could be deleted
Line 194: „number of calories consumed“ should be replaced by „energy intake“
Author Response
Response to Reviewer 1
Dear Reviewer:
Thank you very much for your valuable comments. We have extensively revised the manuscript according to your recommendations. Our point-by-point responses are as follows, along with a clear indication of the location of the revision (highlighted by changing the color in the font).
1. Depending on the statistical test that was used, varying results were detected. In Table 1, it is demonstrated that the percentage of CKD was the lowest in vegans, but the highest in lacto-ovo-vegetarians. The authors should discuss this finding and differentiate that the occurrence of CKD was only low in the lacto-ovo-vegetarian group after adjustment of the data (lines 142). At the moment, the authors only state (in the Abstract, Discussion and Conclusions) that vegetarian diets were negatively associated with CKD, however, this is not completely true.
Response
Thanks for your comments. After carefully inspected, the multivariate logistic regression analysis disclosed that both vegetarian dietswere found to be significantly associated with CKD (vegan OR: 0.87, 95% CI, 0.77 to 0.99,P=0.041; ovo-lacto vegetarian OR: 0.84, 95% CI 0.78–0.90, P<0.001). Moreover, as you pointed, it is demonstrated that the percentage of CKD was the lowest in vegans, but the highest in lacto-ovo-vegetarians in Table 1. To clarify this, we further used stepwise, logistic model to analyze which variables contributed the big difference. After adjusted age, gender, and comorbidities (hypertension, diabetes), the vegan and ovo-lacto vegetarian diet subjects had the lower risk in CKD. The results are shown in below.This is not surprising.The age, gender and comorbidities (hypertension and diabetes) were well-known risk factors in CKD research and contributed this difference.
Model 1 | Model 2 | Model 3 | Model 4 | |
OR (95% CI) | OR (95% CI) | OR (95% CI) | OR (95% CI) | |
Omnivores | 1.0 (reference) | 1.0 (reference) | 1.0 (reference) | 1.0 (reference) |
Vegan | 0.90 (0.82-0.98) | 0.83 (0.76-0.91 | 0.85 (0.74-0.97) | 0.87 (0.77-0.99) |
Ovo-lacto vegetarian | 1.30 (1.23-1.37) | 1.31 (1.24-1.38) | 0.83 (0.77-0.89) | 0.84 (0.78-0.90) |
Age | 1.03 (1.03-1.03) | 1.01 (1.01-1.02) | 1.01 (1.01-1.02) | |
Male vs. Female | 1.43 (1.36-1.49) | 1.27 (1.19-1.36) | 1.23 (1.15-1.31) | |
DM | 1.59 (1.42-1.78) | 1.53 (1.37-1.70) | ||
HTN | 1.48 (1.37-1.60) | 1.41 (1.30-1.52) | ||
Current smoking | 1.36 (1.21-1.53) | |||
Abdominal obesity | 0.99 (0.82-1.19) | |||
Systolic BP (per 10 mmHg) | 1.03 (1.01-1.05) | |||
Low HDL | 1.19 (1.11-1.28) | |||
High TG | 1.10 (1.02-1.19) |
2. Moreover, the number of patients was markedly higher in the omnivorous group (n= 39,068) when compared to the ovo-lacto-vegetarian group (n=11,809) and the vegan group (n=4,236). Please comment on this, does it implicate any limitation for the statistical data analysis?
Response
Thanks for your reminding. In our cohort, the vegan group account for around 7.6% in all subjects, and the ovo-lacto vegetarian group account for 21.4%. The percentages of vegetarian diets were slightly lower while compared with Adventist Health Study-2 in USA (vegan group: 8%, ovo-lacto vegetarian group: 30%). (Nutr Metab Cardiovasc Dis 23: 292-2, 2013). Therefore, we think this is the special character of our cohort rather than a limitation. We had added these statements to our discussion (Page 5, Line 162-169)
3. Minor comments: Line 93: „three groups“ – please define. Line 102: please replace „were“ by „was“. Lines 111/112: please add the age for the omnivorous group
Response
We have revised our manuscript and added sentences in accordance with your suggestions. (Page 3, Line 94) (Page 3, Line 103) (Page 3, Line 113)
4. Minor comments: Line 156: „most of the subjects were Buddhists“ – please explain, why this is a limitation. Line 166: „causes“ – or do the authors mean „risk factors“?
Response
Thanks for your reminding. We revised our sentence and explained why the Buddhists priest is a limitationin Discussion section. ‘’The Buddhist priestslived an ascetic and communal life with little exercise.’’ We correct the word ‘’cause’’ to ‘’risk factors’’. (Page 5, Line 155-156) (Page 6, Line 170)
5. Line 182-183: „In addition, a plant-based diet that focuses on fruits, vegetables and fiber is strongly believed to protect against the development of metabolic syndrome.“ – this aspect is already mentioned in the sentence in Lines 180-182 („…and a decreased risk of metabolic syndrome“) and could be deleted. Line 194: „number of calories consumed“ should be replaced by „energy intake“
Response
Based on your suggestions, we have deleted the sentences in Discussion section (Page 6, Lines 184-185), in our manuscript. We also corrected the words. (Page 6, Lines 196)

Reviewer 2 Report
Your research question is of interest and your central hypothesis is sound. The main issues I identified with the manuscript surround variables in your analysis that appear to be extraneous to your research question and inadequate discussion of your results.
Your research question is only about type of diet. In the first paragraph of your methods you introduce new variables, such as betel nuts, smoking, and drinking alcohol. There is no reason given for why you think these risk behaviors are relevant to your research question about diet.
In line 72 your definition of an alcohol drinking habit has no clinical referent. As described I do not believe it is in line with the clinical definition.
In line 75 you state "we measured waist circumference." As written it implies that you took that measurement yourself which I don't believe to be the case.
In line 93 I think you mean the word "discrete" or "categorical" and not "normal".
With such a big sample, I think you need to explain why you state in your statistical analysis section that you imputed missing values rather than removing those cases from the dataset entirely. Was there a difference in the people with and without missing data?
In line 111 it's odd that you include numerical values for only 2 of the 3 groups.
In Table 1, you give the percentages for distribution of gender and age groups in your three diet groups. I think that row percents would give more useful information than the column percents that you include.
In Table 1, for diabetes it seems that vegans have a higher proportion of diabetes than the other two groups. Do you discuss that at all? Also, remove the < next to 0.042.
In Table 1, you present data about different measures of cholesterol. What is the relationship between these variables and CKD?
In Table 1, the note at the bottom should present information in the same order as the table.
In line 141, do you ever explain why these are the variables you chose to control for in your regression? What is the reasoning for inclusion of each?
In Table 2, a big difference occurs between the univarite and multivariate odds ratio for vegetarians. You do not explain your thoughts on why you think this is happening. Which variables are contributing to this difference? Why are they important?
Figure 2 does not re-present information from Table 2 in a meaningful enough way to justify its inclusion.
Author Response
Response to Reviewer 2
Dear Reviewer:
Thank you very much for your valuable comments. We have extensively revised the manuscript according to your recommendations. Our point-by-point responses are as follows, along with a clear indication of the location of the revision (highlighted by changing the color in the font).
1. The main issues I identified with the manuscript surround variables in your analysis that appear to be extraneous to your research question and inadequate discussion of your results. Your research question is only about type of diet. In the first paragraph of your methods you introduce new variables, such as betel nuts, smoking, and drinking alcohol. There is no reason given for why you think these risk behaviors are relevant to your research question about diet.
Response
Thanks for your comments. Although some variables (betel nuts, smoking, drinking alcohol) were not well-known risk factors for CKD, in some small or local studies showed that their associations with CKD. In Hsu’s study, the betel nuts chewer had higher prevalence of CKD as compared to non-chewer (Nephrology 16(8): 751-7, 2011). Literatures also reported that smoking was associated with an increased risk of CKD (OR 1.63, p=0.02). (BMCPublic Health 1: 731, 2010). Previous studies report conflicting results of a dose-dependent association between alcohol consumption and incidence of CKD. Moreover, alcohol use disorder was associated with increased incident of CKD (PLOS ONE 13(9): e0203410.1, 2018). This is the reason why these three variables were included in our analysis.
2. In line 72 your definition of an alcohol drinking habit has no clinical referent. As described I do not believe it is in line with the clinical definition.
Response
Thanks for your kind remind. The definitions about alcohol drinking varied in many studies or clinical situation. Since our cohort is aim to find the association of chronic disease and vegetarian diets, we use the same definition of alcohol drinking from the large cohort of Adventist Health Study-2(Nutr Metab Cardiovasc Dis 23: 292-2, 2013). In addition, we looked back to our original questionnaire in this study. We found a mistake about description of alcohol drinking. We revised this sentence. ‘’An alcohol drinking intake was defined as drinking of alcohol for at least once in thepast 12 months.’’(Page 2, Lines 73)
3. In line 75 you state "we measured waist circumference." As written it implies that you took that measurement yourself which I don't believe to be the case.
Response
We correct the misleading word and revised the sentence in our manuscript in Methods section. ‘’Our well-trained nurse measured waist circumference (WC) at the mid-level between the lower edge of the rib cage and the iliac crest with the participants in a standing position.’’(Page 2, Lines 75)
4. In line 93 I think you mean the word "discrete" or "categorical" and not "normal".
Response
Thanks for your kind reminding. The word means ‘categorical’ in Line 96. We had revised it.
5. With such a big sample, I think you need to explain why you state in your statistical analysis section that you imputed missing values rather than removing those cases from the dataset entirely. Was there a difference in the people with and without missing data?
Response
The multiple imputation analysis for miss data was widely used for statistical analysisin many large cohort studies (Lancet 392(10156):1403-141, 2018; Cri Care 22(1):278, 2018). Moreover, according to your request, we strictly excluded those with missing data and further analyzed it by multivariate analysis. The results are shown in below. We found that the trend of results were the similar asthe analysis with imputed missing values.
OR (95% CI) | p value | |
Omnivores | 1.0 (reference) | 1.0 (reference) |
Vegan | 0.89 (0.78-1.01) | 0.07 |
Ovo-lacto vegetarian | 0.85 (0.79-0.91) | <0.001 |
6. In line 111 it's odd that you include numerical values for only 2 of the 3 groups.
Response
Based on your suggestions, we have added numerical values of omnivores group in Results section. (Page 3, Lines 113)
7. In Table 1, for diabetes it seems that vegans have a higher proportion of diabetes than the other two groups. Do you discuss that at all? Also, remove the < next to 0.042.
Response
Thanks for your reminding. Actually, previous studies ever showed subjects with vegetarian diets were associated with lower prevalence diabetes as compared to omnivores. Because of the cross-sectional design of in nature, the causal relationship could not be established between vegetarian diets and diabetesin our cohort. Since many subjects in our cohort were self-disciplined Buddhist priests or volunteers, once they were diagnosed with diabetes, they were prone to have more strict diet control with vegan.This is the explanation why vegans had a higher proportion of diabetes in vegan group than the other two groups. In addition, we corrected the error and deleted the < next to 0.042 in Table 1.
8. In Table 1, you present data about different measures of cholesterol. What is the relationship between these variables and CKD?
Response
Thank you for your question. The different measures of cholesterol were defined by Taiwan metabolic syndrome definition. In the past studies, the different diet patterns were associated with metabolic syndrome. (Br J Nutr. 113 Suppl 2:S136-43, 5015) In addition, metabolic syndrome was the risk factor of CKD. (Curr Opin Nephrol Hypertens. 22(2):198-203, 2013) Thus, we present to different measures of cholesterol to analysis the relation with CKD.
9. In Table 1, the note at the bottom should present information in the same order as the table.
Response
Thanks for your reminding. We revised the note at bottom and adjusted the order in Table 1.
10. In line 141, do you ever explain why these are the variables you chose to control for in your regression? What is the reasoning for inclusion of each?
Response
Thanks for your comment. Those variables were both risk factors for CKD and metabolic syndrome. (BMC Nephrology 16: 83, 2015; Arch Intern Med. 163(4):427-436, 2013) The different diets have been proven the relation with metabolic syndrome. Thus, we included those variables to control our regression.
11. In Table 2, a big difference occurs between the univarite and multivariate odds ratio for vegetarians. You do not explain your thoughts on why you think this is happening. Which variables are contributing to this difference? Why are they important?
Response
Thanks for your comment. The characteristics in different diet patterns individuals had the significant differences in age, gender, and diabetes (see table 1). These factors may affect the association in diet patterns and CKD. By the stepwise process, we compared several nested model results of the variables selection. It is helpful to clarify the change the association of the diet patterns and CKD from the crude model to the final model. Therefore, we showed the more simplified models with few variables than our final model. After adjusted age, gender, hypertension and diabetes in the multiple logistic model (Model 3), the vegan and ovo-lacto vegetarian diet subjects became associated with the lower risk in CKD. The results are shown below. This is not surprising that he age, gender and comorbidities of hypertension and diabetes were well-known risk factors in CKD research. In addition, the comorbidity prevalence difference often existed in the diet patterns studies and sometimes confounder the relationship of the diet patterns and outcome diseases. Hence, the diabetes and hypertension contributed to the difference between the univariate and multivariate odds ratio for vegetarians in our analysis.
Model 1 | Model 2 | Model 3 | Model 4 | |
OR (95% CI) | OR (95% CI) | OR (95% CI) | OR (95% CI) | |
Omnivores | 1.0 (reference) | 1.0 (reference) | 1.0 (reference) | 1.0 (reference) |
Vegan | 0.90 (0.82-0.98) | 0.83 (0.76-0.91 | 0.85 (0.74-0.97) | 0.87 (0.77-0.99) |
Ovo-lacto vegetarian | 1.30 (1.23-1.37) | 1.31 (1.24-1.38) | 0.83 (0.77-0.89) | 0.84 (0.78-0.90) |
Age | 1.03 (1.03-1.03) | 1.01 (1.01-1.02) | 1.01 (1.01-1.02) | |
Male vs. Female | 1.43 (1.36-1.49) | 1.27 (1.19-1.36) | 1.23 (1.15-1.31) | |
DM | 1.59 (1.42-1.78) | 1.53 (1.37-1.70) | ||
HTN | 1.48 (1.37-1.60) | 1.41 (1.30-1.52) | ||
Current smoking | 1.36 (1.21-1.53) | |||
Abdominal obesity | 0.99 (0.82-1.19) | |||
Systolic BP (per 10 mmHg) | 1.03 (1.01-1.05) | |||
Low HDL | 1.19 (1.11-1.28) | |||
High TG | 1.10 (1.02-1.19) |
12. Figure 2 does not re-present information from Table 2 in a meaningful enough way to justify its inclusion.
Thank you for your suggestions. We deleted Figure 2 and revised our sentence as your request.

Round 2
Reviewer 1 Report
Thank you for considering my comments.
A few minor comments:
- Line 126: “NAFLD” – abbreviation/parameter not presented in the Table
- Line 164: suggest to write “accounted”
- Lines 166/167: Please write “This may BE another special character of our cohort in Taiwan.”
Author Response
Response to Reviewer 1 round 2
Dear Reviewer:
Thank you very much for your valuable comments. We have extensively revised the manuscript according to your recommendations. Our point-by-point responses are as follows, along with a clear indication of the location of the revision (highlighted by changing the color in the font).
1. Line 126: “NAFLD” – abbreviation/parameter not presented in the Table
Response
Thanks for your reminding, we have deleted the word in Table 1.
2. Line 164: suggest to write “accounted”
Response
Thanks for your suggestion, we have rewritten the sentence. (Page 6, Line 177)
3. Lines 166/167: Please write “This may BE another special character of our cohort in Taiwan.”
Response
Based on your suggestions, we have revised the sentence.(Page 6, Line 181)

Reviewer 2 Report
Authors,
I appreciate the time you took to revise your manuscript and respond to my comments. However, a majority of your response was in the letter to me and not in edits to the manuscript. My comments were about missing information and questions in the manuscript, and the point was for you to address the comments in the manuscript.
Author Response
Response to Reviewer 2 round 2
Dear Reviewer:
Thank you very much for your valuable comments. We have extensively revised the manuscript according to your recommendations. Our point-by-point responses are as follows, along with a clear indication of the location of the revision (highlighted by changing the color in the font).
1. However, a majority of your response was in the letter to me and not in edits to the manuscript. My comments were about missing information and questions in the manuscript, and the point was for you to address the comments in the manuscript.
Response
Based on your suggestions, we have revised our manuscript and added sentence in the Materials and Methods, Result section and Discussion section.(Page 2, Line 73) (Page 3, Line 99-100) (Page 3, Line 102-105) (new Table 2) (Page 5, Line 139-152) (Page 6, Line 168-174) (Page 6, Line 195-201) (Page 7, Line 210-217)

Round 3
Reviewer 2 Report
Thank you for an excellent second round of revisions. There are only a few minor issues left to address.
1. Your research question (page 2, lines 51-53) still is much more narrow than the analysis you performed. It is not appropriate to include risk behaviors given your current research question. You need to either expand your research question to include risk behaviors or remove them from your analysis.
2. If you include risk behaviors, a shorter version of the paragraph in the discussion (page 6, lines 209-216) needs to be moved to the methods. In the discussion is too late.
3. You need to take another pass with an English language editor for your new edits. I noticed many errors compared to the high quality of the rest of the manuscript.
Author Response
Response to Reviewer 2 round 3
Dear Reviewer:
Thank you very much for your valuable comments. We have extensively revised the manuscript according to your recommendations. Our point-by-point responses are as follows, along with a clear indication of the location of the revision (highlighted by changing the color in the font).
1. Your research question (page 2, lines 51-53) still is much more narrow than the analysis you performed. It is not appropriate to include risk behaviors given your current research question. You need to either expand your research question to include risk behaviors or remove them from your analysis.
Response
Based on your suggestions, we have deleted risk behaviors and revised our manuscript and in the Materials and Methods, Result section and Discussion section.
2. If you include risk behaviors, a shorter version of the paragraph in the discussion (page 6, lines 209-216) needs to be moved to the methods. In the discussion is too late.
Response
Thanks for your suggestion, we have deleted risk behaviors and revised our manuscript.
3. You need to take another pass with an English language editor for your new edits. I noticed many errors compared to the high quality of the rest of the manuscript.
Response
Thanks for your kindly reminding, we have sent our manuscript for English editing and revised our sentences.
